# *Escherichia coli* Activate Extraintestinal Antibody Response and Provide Anti-Infective Immunity

**DOI:** 10.3390/ijms25137450

**Published:** 2024-07-07

**Authors:** Xiang Liu, Xuanxian Peng, Hui Li

**Affiliations:** State Key Laboratory of Bio-Control, School of Life Sciences, Southern Marine Science and Engineering Guangdong Laboratory (Zhuhai), Guangdong Key Laboratory of Pharmaceutical Functional Genes, Sun Yat-sen University, Guangzhou 510275, China; liuxiang888525@163.com

**Keywords:** intestinal microflora, extraintestinal, acquired immunity, antibody, *Escherichia coli*, outer membrane proteins

## Abstract

The effects of intestinal microflora on extraintestinal immune response by intestinal cytokines and metabolites have been documented, but whether intestinal microbes stimulate serum antibody generation is unknown. Here, serum antibodies against 69 outer membrane proteins of *Escherichia coli*, a dominant bacterium in the human intestine, are detected in 141 healthy individuals of varying ages. Antibodies against *E. coli* outer membrane proteins are determined in all serum samples tested, and frequencies of antibodies to five outer membrane proteins (OmpA, OmpX, TsX, HlpA, and FepA) are close to 100%. Serum antibodies against *E. coli* outer membrane proteins are further validated by Western blot and bacterial pull-down. Moreover, the present study shows that OstA, HlpA, Tsx, NlpB, OmpC, YfcU, and OmpA provide specific immune protection against pathogenic *E. coli*, while HlpA and OmpA also exhibit cross-protection against *Staphylococcus aureus* infection. These finding indicate that intestinal *E. coli* activate extraintestinal antibody responses and provide anti-infective immunity.

## 1. Introduction

Throughout our life, we provide a residence to hundreds of individual bacterial species comprising numerous microbial communities, referred to as the human microbiota [1]. The human microbiota establishes a symbiotic relationship with the host and helps maintain physiological homeostasis due to having positively active roles in our nutrients and immunities [2], playing an important role in the balance between our health and disease [3,4]. Hence, host health is paramount to the microbiota.

Human microbial colonization occurs in the skin, genitourinary system and, mainly, in the oral cavity and gastrointestinal tract [2]. Microbes in the gastrointestinal tract construct gut microbiota that is a commensalism of microorganisms. The gut microbiota not only helps provide essential nutrients to us by metabolizing proteins and complex carbohydrates and synthesizing vitamins but also stimulates hosts to generate innate and acquired immune responses by producing an enormous number of metabolic products [5,6]. The bowel establishes a balanced environment between pathogens and normal microflora and has multiple defenses, including antimicrobial peptides, the secretory IgA (sIgA) mucus layer, and the epithelial layer [7,8]. When the balance of the environment is disturbed, gut microbiota dysbiosis results [9,10]. This may be caused by fecal transplantation and combatted by drinks such as beer [11]. More importantly, recent studies have shown that gut barrier dysfunction causes changes in the composition and function of the gut microbiota [12], which has led to the development of the concept of leaky gut syndrome [13]. Leaky gut syndrome contributes to autoimmune diseases such as type 1 diabetes, multiple sclerosis, rheumatoid arthritis, and celiac disease [14]. This is because the damaged intestinal barrier and the following increase of intestinal permeability allow antigens to pass more easily and activate the immune system or cross-react with extraintestinal tissues, respectively [15]. There are evidences that intestinal microflora disorder affects extraintestinal diseases through the “gut–lung” axis and the “gut–brain” axis [16,17,18]. These findings suggest that protein antigens from intestinal microflora may enter extraintestinal tissues and activate abnormal acquired immune responses, causing autoimmune diseases. However, whether the normal acquired immune response is activated and plays a role in anti-infective immunity is unknown. 

To address this question, it needs to be clear that microbes stimulating acquired immune response should settle down in the intestine of all human beings but do not enter and colonize in the extraintestinal tissues as members of the normal flora. Ideally, they exist in all healthy individuals. Furthermore, serum antibody count should be the most suitable marker due to its specificity. In addition, serum antibodies can protect hosts from infection caused by microbes. 

*Escherichia coli* is a dominant bacterium in the diverse commensal microbiota that does not live in extraintestinal tissues as one of normal bacterial flora. As one among the first bacterial species to colonize the intestine of a newborn, it reaches a very high density of 10^8^ CFU per gram of feces [19,20]. Nonpathogenic *E. coli* do provide health benefits to the host by improving the microbial balance of intestinal tract and the synthesis of vitamin K [21,22]. Therefore, *E. coli* could be used as a bacterial representative to explore whether intestinal microflora stimulates the production of serum antibodies and its significance for further understanding the role of intestinal microflora. 

Here, *E. coli* was used as a representative bacterium of intestinal microflora to explore whether there were antibodies against *E. coli* outer membrane proteins in the serum of healthy individuals. These antibodies played a role in anti-infective immunity for understanding the acquired immune response induced by intestinal microflora. 

## 2. Results

### 2.1. Preparing of E. coli Outer Membrane Protein Array Chip

Outer membrane proteins are located in the outer membrane of Gram-negative bacteria, including *E. coli*, and are easily recognized by the immune system to stimulate antibody production [23,24,25]. These motivated us to detect serum antibodies against outer membrane proteins for the purpose of exploring the immune response to *E. coli*. To do this, a protein array chip with *E. coli* outer membrane proteins was prepared. Among 82 genes encoding *E. coli* outer membrane proteins annotated in GenBank, 69 have been cloned to pET28a or pET32a expression vector for recombinant proteins (stored in our laboratory). Their information is summarized in Appendix A. These recombinant outer membrane proteins were expressed in *E. coli* BL21 and isolated using His-tag affinity or inclusion body isolation. All recombinant outer membrane proteins were electrophoretically pure (Figure 1A). The resulting recombinant outer membrane proteins were separately spotted on NC membranes for the protein array chip, with bovine serum albumin (BSA) and Tris-HCl buffer used as negative controls and supernatant of *E. coli* BW25113 lysates used as a positive control (Figure 1B). 

### 2.2. Prevalence of Serum Antibodies against E. coli Outer Membrane Proteins

The prepared protein array chips were used to detect human serum antibodies against *E. coli* outer membrane proteins. To do this, sera from 141 healthy individuals at different stages of age were analyzed (Table 1). They were grouped as newborn (0 y, umbilical cord blood), young children (1–6 y), children (7–14 y), youth (15–50 y), and older (51–83 y) as described previously [26,27]. Figure 2A shows examples of immune reactions between sera from different stages of age and outer membrane proteins in the microarrays. As demonstrated in Table 2, the prevalence of the antibodies to *E. coli* outer membrane proteins was widely distributed, which was mostly related to types of outer membrane proteins. Specifically, newborn, young children, children, youth, and older had a similar frequency of serum antibodies against the 69 outer membrane proteins, ranking 25.39 ± 31.00, 22.13 ± 29.80, 28.80 ± 30.96, 23.50 ± 27.95 and 25.15 ± 29.06, respectively. However, a differential immune reaction to the 69 outer membrane proteins was detected. Among the 69 outer membrane proteins, 10 and 22 stimulated no and few antibodies (<10%), respectively; five (FepA, HlpA, OmpA, OmpX, and TsX) generated high frequency of antibodies with mean ± SD 82.98 ± 11.63%, 90.07 ± 11.2%, 97.87 ± 2.02%, 93.62 ± 7.69%, and 92.91 ± 7.18%, respectively; and the remaining proteins had middle frequency of antibodies (Table 2). Furthermore, Western blot was used to validate the immune reaction between the outer membrane proteins and the sera. To do this, one serum that contained antibodies against FepA, Tsx, OmpX, or/and HlpA in each stage of age of human individuals was used as the primary antibody. These five sera were separately reacted with a mixture of four outer membrane proteins that was transferred to the NC membrane. Four bands that were located at the corresponding molecular masses were detected. Specifically, OmpX and HlpA were recognized by the serum from newborn, FepA, OmpX, and HlpA were recognized by the serum from young children, Tsx, OmpX, and HlpA were recognized by the serum from children, FepA and HlpA were recognized by the serum from youth, and FepA and OmpX were recognized by the serum from older (Figure 2B). These are consistent with the results in arrays. Furthermore, two sera containing high titer of antibodies against FepA, OmpA, Tsx, OmpX, and HlpA in each stage of age were used as the primary antibodies to react with whole-cell lysates that were transferred to the NC membrane. Corresponding bands were detected in these groups with different stages of age (Figure 2C). Finally, cluster analysis was carried out using R software to understand the relationship between the frequency and different stages of age and outer membrane proteins. For the relationship between the frequency and ages, the analysis showed that newborn, children, youth, and older were separately clustered, and that the four were clustered with young children (Figure 2D). For the relationship between the frequency and outer membrane proteins, three big clusters were determined. In detail, middle, low, and high frequency of antibodies were located in clusters 1 (22 outer membrane proteins), 2 (37 outer membrane proteins), and 3 (10 outer membrane proteins), respectively (Figure 2D). These results indicate that antibodies against *E. coli* outer membrane proteins are widely present in serum in healthy individuals of different ages, but only certain outer membrane proteins stimulate high frequency of antibodies.

### 2.3. Titer of Serum Antibodies against High Frequency of Outer Membrane Proteins

As described above, antibodies against OmpA, OmpX, TsX, HlpA, and FepA had higher frequency than the others and thereby were selected for measurement of the titer. Among the 141 serum samples, most had the titers with 1:250–1:2000 against FepA, HlpA, OmpX, Tsx, and with 1:1000–1:2000 against OmpA. The top titers were located at 1:500 (26 samples), 1:1000 (27 samples), 1:2000 (43 samples), 1:1000 (32 samples), and 1:500 (31 samples) against FepA, HlpA, OmpA, OmpX, and Tsx, respectively. Interestingly, when the serum was diluted at 1:3000 or higher, few cases (less than four cases) were positive in antibodies against FepA, HlpA, OmpX, and Tsx, but 15 cases were positive in antibodies to OmpA (Figure 3A). Cluster analysis showed a weak correlation between these five antibodies (Figure 3B). Note that this cluster analysis is not consistent with the above cluster (Figure 2D) for these five antibodies due to the basis of antibody title instead of age. These results indicate that these outer membrane proteins with a high frequency of antibodies stimulate high antibody titers.

### 2.4. Titers of Serum Antibodies against E. coli Outer Membrane Proteins

Surfaced-exposed outer membrane proteins are crucial for bacterial–host interaction [28]. Thus, we could use *E. coli* cells to react with serum directly for confirmation of these specific antibodies to *E. coli*. To do this, suitable serum dilution was tested. When 10^8^ colony forming units (CFUs) of formaldehyde-inactivated *E. coli* were incubated with 50 pooled sera diluted from 1:0 (no dilution), 1:10–1:1000 of bovine serum albumin (BSA) and PBS were used as negative controls. After reaction with HRP-anti-human IgG, OD450 nm was measured. The optical density was proportional to the dilution during 1:10–1:1000 but lower in 1:0 than 1:10. Approximately 2.1, 3.0, 1.8, and 0.5 of OD value was measured at 1:0, 1:10, 1:100, and 1:1000 dilution of serum samples, respectively, when about 0.10 of OD value was detected in the negative controls (Figure 4A). Thus 1:100 dilutions were adopted to test 50 samples from different stages of age, ten for each stage. OD values (Mean ± SD) 1.711 ± 0.449, 2.014 ± 0.434, 1.925 ± 0.0.545, 2.132 ± 0.54, and 2.28 ± 0.496 were characterized in 0, 1–6 y, 7–14 y, 15–50 y, and 50–83 y, respectively (Figure 4B). These results indicate that antibodies against *E. coli* were rich in human sera.

### 2.5. Total Serum Antibodies against E. coli Outer Membrane Proteins 

To quantify serum antibody to *E. coli*, pooled sera were treated by 40% saturated ammonium sulfate for immunoglobulins. Then, bacterial pull-down was used to measure serum antibody binding with *E. coli* by mixing the immunoglobulins with formaldehyde-inactivated *E. coli*. These bound antibodies were isolated by glycine-HCl buffer, then precipitated by acetone. The pellets were dissolved in phosphate buffered saline (PBS) and isolated by SDS-PAGE. Three bands with approximate molecular masses of 70, 54, and 25, corresponding to IgM heavy chain, IgG heavy chain, and IgM and IgG light chain, respectively, were dominant (Figure 5). Using quantified BSA as a standard of protein abundance (Appendix A), percent binding of purified serum immunoglobulins with *E. coli* was calculated. When 36 mg immunoglobulins were mixed with 10^11^ CFU of *E. coli*, approximately 21 μg of bound antibodies was detected (Appendix A). Therefore, 0.057% serum immunoglobulins reacted with *E. coli*, supporting the conclusion that there is a specific antibody against *E. coli* in the blood of healthy individuals. 

### 2.6. Active Immune Protection Induced by Outer Membrane Proteins with the High and Middle Frequencies of Antibodies

To explore the ability in immune protection of outer membrane proteins against bacterial infection, 22 outer membrane proteins were separately used to immunize mice. The immunized mice were challenged with virulent *E. coli*. Among the mice immunized by these outer membrane proteins, seven exhibited immune protection but the others did not. Specifically, relative percent survival (RPS) of mice immunized by the seven outer membrane proteins ranked from high to low was OstA (81.82%) > HlpA, Tsx, NlpB, OmpC, YfcU (63.64%) > OmpA (54.54%) (Table 3). Furthermore, HlpA, OmpA, and Tsx were selected to test whether these outer membrane proteins stimulated cross-immune protection against infection caused by *S. aureus*. The cross-immune protection was detected in mice immunized by HlpA and OmpA but not Tsx (Table 4). These results indicate that the antibody immune response stimulated by outer membrane proteins protects mice from infection caused by Gram-negative and -positive bacteria. 

To understand the cross-immune protection, phylogenetic trees were performed based amino acid sequence of OmpA and HlpA. OmpA of *E. coli* and *S. aureus* was clustered together, while HlpA of *E. coli* and *S. aureus* had also a relation (Appendix A), suggesting that there is a homology between the two OmpA and HlpA. 

## 3. Discussion

Intestinal microflora influences extraintestinal immune response by intestinal cytokines and metabolites has been documented [29,30], but whether protein antigens from the intestinal microflora may stimulate an extraintestinally acquired immune response is unknown. The present study explores the possibility through measuring serum antibody against *E. coli* outer membrane proteins in different age stages of healthy individuals and identifying antibody protection against bacterial infection. *E. coli* is a versatile species which comprises harmless commensals and different pathogenic variants. Most strains are harmless commensals that live in a mutually beneficial association with their hosts, while some strains either cause intestinal or extraintestinal diseases in humans and many animal hosts, which are referred to as pathogenic variants [31,32]. The pathogenic variants are classified as intestinal pathogenic *E. coli* (IPEC) and extraintestinal pathogenic *E. coli* (ExPEC). IPEC cause a variety of diarrheal illnesses and extraintestinal syndromes such as hemolytic-uremic syndrome, while ExPEC cause urinary tract infections, bloodstream infection, sepsis, and neonatal meningitis [33,34]. IPEC have their characteristic virulence factors responsible for their associated clinical manifestations, which can be distinguished from commensal *E. coli* [33]. Because of the ability in invading extraintestinal tissues [35,36,37], IPEC and ExPEC can activate immune system to generate acquired immunity [38,39]. However, not all individuals are infected by IPEC and ExPEC. The present study shows that serum antibodies against *E. coli* outer membrane proteins are detected in all healthy individuals tested, some of which protect hosts from bacterial infection in a mouse model. Therefore, *E. coli* as a bacterium of intestinal microflora can stimulate extraintestinal acquired immunity to protect hosts from bacterial infection. The finding provides a new insight into the role of the intestinal microflora. 

Outer membrane proteins of Gram-negative bacteria are key molecules that interface the cells with the physical, chemical, and biological environment [40]. Because of being located in the outmost of the cells, they are easily recognized by host immune system. This is also a reason why identification of immunogens from outer membrane proteins as vaccine candidates is highly recommended [41,42]. Therefore, serum antibodies to outer membrane proteins are a reasonable proof to understand the effect of intestinal microflora on extraintestinal acquired immunity. The present study carefully examines antibodies to *E. coli* outer membrane proteins in the sera of healthy individuals to support the proposal that the intestinal microflora activates extraintestinal acquired immunity. To explore this, these serum antibodies were validated by qualitative and quantitative tests using protein array, enzyme-linked immunosorbent assay (ELISA), and bacterial pull-down. The results of negative staining on serum antibodies to two-component system and some outer membrane proteins in contrast with positive staining on serum antibodies to the other outer membrane proteins supports the specificity of the array assay. The findings that antibodies against outer membrane proteins are detected in all healthy individuals tested and frequencies of five antibodies to OmpA, OmpX, TsX, HlpA, and FepA are close to 100%, as well as 0.057% serum immunoglobulins are reacted with *E. coli*, indicate that these extraintestinal acquired immune response is common. Ahlstedt et al. tested the effect of ingestion of probiotic *E. coli* O83 by adults on serum antibody response. They found that the response was detected in four out of fourteen cases examined, which they attributed to a transient irregular colonization of these bacteria [43], suggesting that intestinal *E. coli* activates serum antibody generation. On the other hand, serum antibodies to *E. coli* are generated following pathogenic *E. coli* infection that causes enteritis, urinary tract infection, septicaemia, and other clinical infections [35,36,37]. However, it is impossible for all healthy individuals to have experience with invisible infections, especially young children, whose antibodies should come from their mothers as newborns (0 years) [44]. Importantly, the detection of 69 antibodies against *E. coli* outer membrane proteins is key to prove this possibility. Therefore, these serum antibodies should be a response to intestinal *E. coli* antigens. However, how these antigens activate extraintestinal acquired immunity awaits investigation. This is also related to why are there differential antibody responses to these outer membrane proteins. For example, antibodies to OmpA and OmpC display approximately 100% and 0%, respectively. Presumably, this is related to the differential exposure of these outer membrane proteins to the extraintestinal immune system and the strength of immunogenicity of these outer membrane proteins. Furthermore, the positions of outer membrane proteins and the relative amount of outer membrane protein produced may also be responsible for it. Thus, these outer membrane proteins are differentially recognized by the extraintestinal immune system. 

Another key question is whether the antibodies provide immune protection against bacterial infection. The present study answers this question through specific immune protection and cross-immune protection. Seven outer membrane proteins (OstA, HlpA, Tsx, NlpB, OmpC, YfcU, and OmpA) with high frequency in the sera of healthy individuals show a specific immune protection of 54.54–81.82% against pathogenic *E. coli* infection, while HlpA and OmpA exhibit cross-protection of 40.74% and 37.04% against *S. aureus* infection. Among the seven outer membrane proteins, reports have indicated that OmpA from not only *E. coli* but also *Pichia mirabilis*, *Vibrio ichthyoenteri*, and *Brucella abortus* show specific immune protection [45,46,47,48,49]. Meanwhile, OmpA from *V. alginolyticus* has abilities to fight against infections not only caused by *V. alginolyticus* but also by *Edwardsiella tarda*, *Aeromonas hydrophila,* and *Pseudomonas fluorescens*, suggesting cross-immune protection [50,51]. More interestingly, the antibody cross-reactivity is also detected between infectious Gram-negative and –positive bacteria [52,53], although differences in outer cell envelopes exist between Gram-positive and Gram-negative bacteria [54]. The present study further shows that *E. coli* OmpA and HlpA provide cross-protection against *S. aureus* infection, which is related to the homologies between *E. coli* OmpA and HlpA and *S. aureus* OmpA and HlpA, respectively. Therefore, the homology of amino acid sequences provides an explanation for the cross-protection. 

## 4. Materials and Methods

### 4.1. Ethics Statement

Human and animal protocols were approved by the University Committee for the Use and Care of Laboratory Animals. Human sera were randomly obtained from healthy physical examination normal individuals and umbilical cord blood was randomly collected from full term natural delivery of healthy maternal in Zhongshan Hospital, Xiamen, China. Sera were isolated by anticoagulant and kept at −80 °C until use. The blood collection was conducted according to the approval of the Ethics Committee of the hospital for use of serum samples in clinical chemistry. All animal work was conducted in strict accordance with the recommendations of the Guide for the Care and Use of Laboratory Animals of the National Institutes of Health. The protocol was approved by the Institutional Animal Care and Use Committee of Sun Yat-sen University (approval no. SYSU-IACUC-2020-B126716).

### 4.2. Materials

Bacterial strains and *E. coli* outer membrane protein recombinant plasmid were preserved in our laboratory, including *E. coli* K12 BW25113, *E. coli* Y17, S. aureus. Sera of healthy individuals were obtained from Zhongshan Hospital, Xiamen University, China. Kunming mice were provided from laboratory animal center of Sun Yat-sen University (China). Nitrocellulose (NC) membrane was purchased from Millipore (Ireland). Ni-NTA resin was taken from Qiagen (Dusseldorf, Germany). Other chemicals were all analytical reagents.

### 4.3. Protein Expression and Purification

Recombinant proteins were purified by affinity chromatography using Ni-NTA resin (Qiagen) or isolation of inclusion bodies. Briefly, *E. coli* BL21 harboring outer membrane recombinant plasmids were cultured in 600 mL of LB, induced by 0.1 mM IPTG. After 8 h induction at 37 °C, cells were harvested, washed, resuspended in buffer A (50 mM Tris-HCl, pH 7.4), and lysed by sonication. Following centrifugation at 5000× *g* for 10 min, the supernatant and the pellet were collected for identification of recombinant proteins by SDS-PAGE. The recombinant proteins in supernatant were purified by Ni-NTA resin using stepwise against increasing concentrations of imidazole containing buffer A. The imidazole was removed by dialysis in buffer A solution. The recombinant proteins in the pellet were purified by isolation of inclusion bodies. The isolation was performed by washing using buffer B (50 mM Tris, 5 mM EDTA, 1 M NaCl, 0.05% Triton X-100, 5% glycerol, pH 8.0) and buffer C (50 mM Tris-HCl, 5 mM EDTA, 100 mM NaCl, 0.05% Triton X-100, 5% glycerol, pH 8.0). These proteins were washed three times by the two buffers. The purified proteins were stored at −80 °C for use.

### 4.4. Protein Chip Preparing and Detection of IgG against Outer Membrane Proteins

The purified outer membrane proteins were diluted to concentrations of 0.25 μg/μL, and a 2 μL solution was arrayed in a 0.7 × 0.7 cm NC membrane. The array consisted of an 8 × 10 matrix with 80 spots, including 69 outer membrane protein spots; the other 11 spots were used as negative and positive controls. After spotting, the NC membrane was blocked with 5% skim milk in TNT buffer (1.211 g Tris, 8.77 g NaCl, and 500 μL Tween-20 in 1 L TNT, pH 7.0) at room temperature for 2 h. The NC membranes were incubated with serum sample at a dilution of 1:500 in TNT buffer containing 5% skim milk at 37 °C for 40 min, rinsed and washed 3 times for 10 min with TNT buffer. Then, the NC membranes were incubated with horseradish peroxidase goat anti-human-IgG at a dilution of 1:1000 in TNT buffer containing 5% skim milk at 37 °C for 1 h. The membrane was washed with TNT buffer and protein spots were detected by DAB, until optimum color developed [22]. When spots were stained, the samples were identified to be positive. Otherwise, the samples were evaluated to be negative. 

### 4.5. Western Blot for Interaction between E. coli and Serum Antibodies

Outer membrane proteins and *E. coli* BW25113 cell lysates were isolated by SDS-PAGE electrophoresis, then transferred to NC membranes for 1 h at 80 V. After the NC membranes were blocked with 5% skim milk, the human serum (1:1000 dilution) was added for 1 h at 37 °C. Then, HRP goat anti-human antibody was added to the NC membrane to incubate at 37 °C for 1 h. After washing 3 times, the protein bands were visualized using a diaminobenzidine (DAB) substrate system.

### 4.6. Bacterial Pull-Down for Binding Efficiency of E. coli with Human Serum Antibody

Human serum immunoglobulins were purified by saturated ammonium sulfate. Briefly, 80 mL pooled human sera were dissolved to PBS with 1:1 (V:V) and saturated ammonium sulfate was slowly added with stirring until the ammonium sulfate reached 40% saturation at 4 °C for 30 min. After centrifugation at 3000 r/min for 10 min, the pellets were collected and re-suspended with PBS. Then, the procedure was repeated. The pellets were dissolved in PBS and dialyzed in PBS overnight to remove the ammonium sulfate. PBS was added to arrive at 80 mL, designated as immunoglobulins. Meanwhile, *E. coli* BW25113 was cultured to OD600 nm = 1.0 and collected at 8000 r/min for 2 min. After washing with 0.85% saline solution, the cells were re-suspended in 1% formaldehyde normal saline solution to inactivate the cells at 80 °C for 90 min. Then, the cells were centrifuged at 8000 r/min for 2 min to remove the formaldehyde solution, and re-suspended with 0.1 M Gly-HCl (pH 2.4) for 10 min. After washing with saline solution twice, the cells were adjusted to OD600 nm = 1.0 and divided into 1.5 mL tubes with equal amount. The cells were collected by centrifugation. Aliquots of 32, 24, 16, and 8 mL purified immunoglobulins were mixed with 1 × 10^11^ CFU *E. coli* and PBS was used as negative control at room temperature for 90 min. After washing, the cells and immunoglobulins were dissociated with 1.2 mL of 0.1 M Gly-HCl (pH 2.4) at room temperature for 10 min. The supernatant was precipitated overnight at −20 °C with acetone and dissolved in 60 μL PBS, then used for SDS-PAGE analysis.

### 4.7. ELISA for Serum Antibodies Binding with E. coli

*E. coli* BW25113 was cultured to OD600 nm = 1.0 and collected by centrifugation at 8000 r/min for 3 min. After washing with 0.85% saline solution, the cells were resuspended with 1% formaldehyde normal saline solution and inactivated at 80 °C for 90 min. The cells were adjusted until OD600 nm = 0.2 and subpackaged to 1 mL/tube (bacterial amount 10^8^ CFU). Then, 100 µL human serum (1:100 dilution) was added to the bacteria at room temperature for 90 min and 100 µL BSA (2 µg/µL) and 100 µL saline solution were used as negative controls. After washing 3 times, 200 µL HRP goat anti-human antibody (1:3000 dilution) was added at room temperature for 1 h. The cells were transferred onto an enzyme-linked plate with 20 µL PBS, and 100 µL coloration liquid (50 µL H_2_O_2_ and 50 µL TMB) was added at 37 °C for 10 min. Finally, 50 µL termination solution (2 M H_2_SO_4_) was added, and the absorbance at OD450 nm was assessed with a microplate reader.

### 4.8. ELISA for Titer of Serum Antibodies against Outer Membrane Proteins

Desirable outer membrane proteins were separately coated in enzyme-linked plate and sealed with skim milk (5%). Human sera were diluted with a desirable dilution and added for incubation. After washing, 200 µL HRP goat anti-human antibody (1:3000 dilution) was added at room temperature for 1 h. Then, the wells had 100 µL coloration liquid (50 µL H_2_O_2_ and 50 µL TMB) added at 37 °C for 10 min. Finally, 50 µL termination solution (2 M H_2_SO_4_) was added, and the absorbance at OD450 nm was assessed with a microplate reader.

### 4.9. Immune Protection of Outer Membrane Proteins against Pathogenic E. coli and S. aureus in a Mouse Model

Immune protection of outer membrane proteins was performed in a mouse model. Briefly, 3-week-old mice were divided into 27 groups (fifteen mice per group) and immunized with 100 μg of outer membrane proteins dissolved in 100 μL of 50 mM Tris-HCl per mouse in complete Freund’s adjuvant. Two booster injections were carried out with the same buffer in incomplete Freund’s adjuvant. One week later, these mice were challenged by intraperitoneal inoculation of 1.5 × 10^8^ CFU *E. coli* Y17 (23 groups) or 1 × 10^11^ CFU of *S. aureus* (4 groups). Then, these challenged mice were observed and recorded daily for two weeks.

### 4.10. Construction of Phylogenetic Tree

The protein sequences of OmpA and HlpA were searched and downloaded through NCBI, then placed in a Fasta format file. Then, a phylogenetic tree of the strains was constructed with MEGA5.02 software based on the methods of Bootstrap test of physiology and Neighbor Joining.

## 5. Conclusions

The present study shows that antibodies against *E. coli* outer membrane proteins are detected in all healthy individuals tested, and demonstrates that some antibodies with high abundance exhibit specific and cross-immune protection. These findings indicate that intestinal *E. coli* activate an extraintestinal acquired antibody response, which highlights the way towards understanding natural anti-infective immunity. 

## Figures and Tables

**Figure 1 ijms-25-07450-f001:**
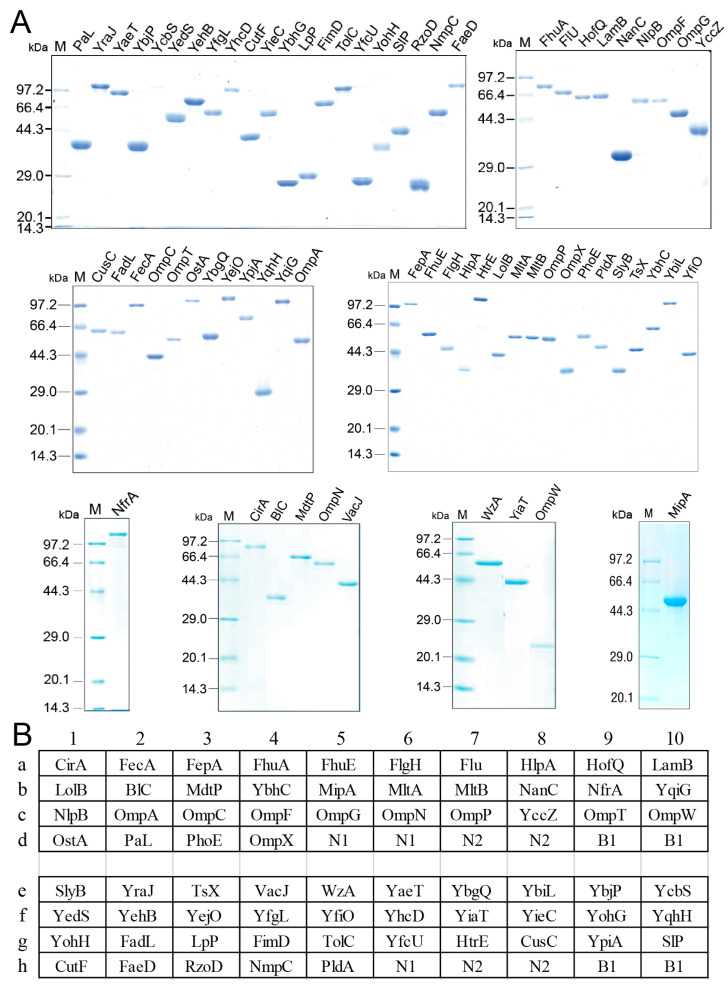
Purity (**A**) and location (**B**) of outer membrane proteins on the microarray. N1 and N2, BSA and 50 mM Tris-HCl, respectively, as negative controls; B1, 0.5 μg supernatant of *E. coli* BW25113 lysates solution as a positive control.

**Figure 2 ijms-25-07450-f002:**
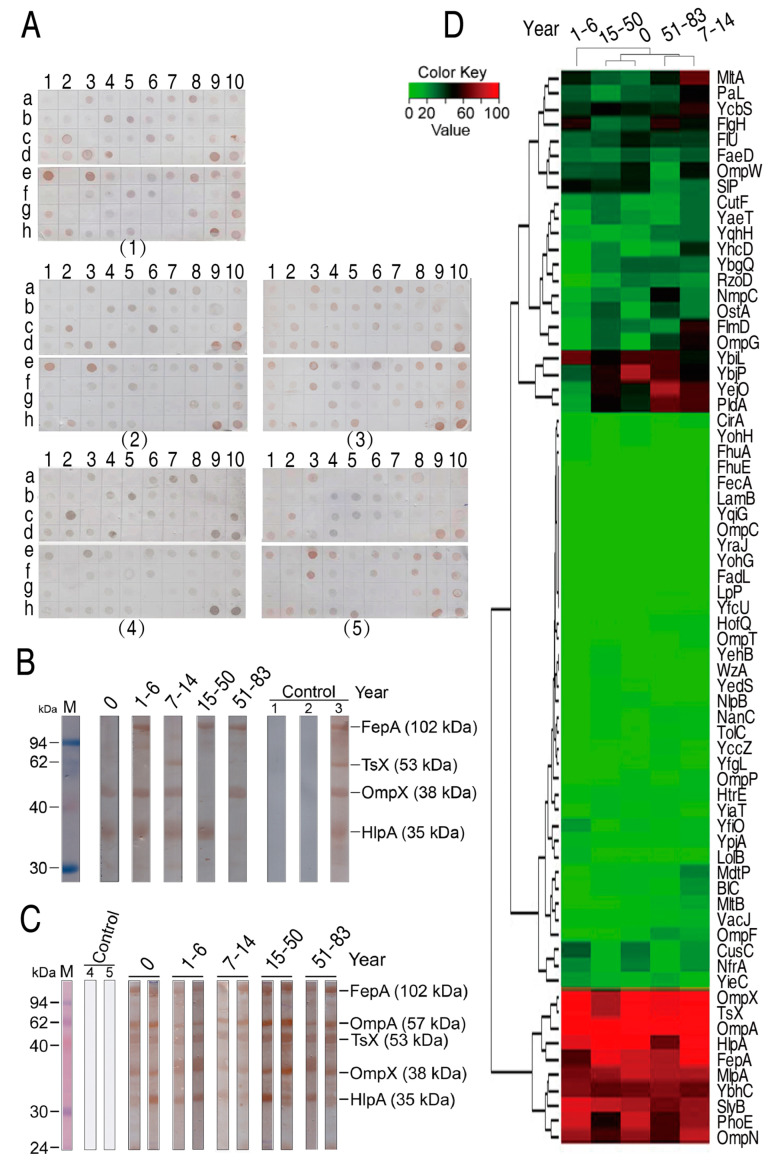
Immune reaction between 69 outer membrane proteins on protein arrays and 141 sera from healthy individuals of varying ages. (**A**) Examples of the immune reaction in 0 y (1), 1–6 y (2), 7–14 y (3), 15–50 y (4), and 51–83 y (5). (**B**,**C**) Western blot for validation of the immune reaction between the indicated outer membrane proteins and the sera from different stages of age. One sera against the indicated outer membrane proteins for each age was selected and a total of five sera (0: No. 20; 1–6: No. 35; 7–14: No. 68; 15–50: No. 85; 51–83: No. 125; Control 1: only serum; Control 2: only anti-human IgG; Control 3: serum 135) were used (**B**); two serum against at least two of the indicated outer membrane proteins were selected for each age (0: No. 22, No. 28; 1–6: No. 40, No. 45; 7–14: No. No. 65, No. 80; 15–50: No. 98, No. 109; 51–83: No. 130, No. 135; Control 4: only Tris-HCl solution; Control 5: only BSA solution) and a total of ten sera were used (**C**). 1 and 2, the secondary antibody horseradish peroxidase (HRP) anti-human IgG was added and HRP anti-human IgG was replaced with anti-human IgG, respectively, as negative controls. 3, positive control; the five pooled positive sera from the serum selected from each age as the primary antibodies. 4 and 5, Tris-HCl and BSA, respectively, were replaced with serum as negative controls. (**D**) Cluster analysis between the frequency and different stages of age and outer membrane proteins.

**Figure 3 ijms-25-07450-f003:**
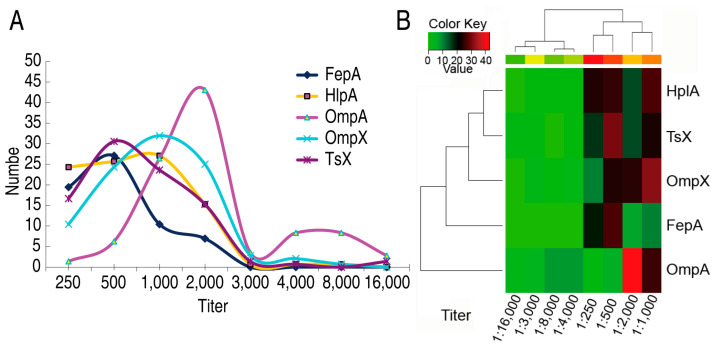
Cluster analysis of immune reaction to outer membrane proteins and antibody titer of high immune reaction sera in different stages of age. (**A**) Number of antibody titer against the indicated outer membrane proteins in different stages of age. (**B**) Cluster analysis of data (**A**).

**Figure 4 ijms-25-07450-f004:**
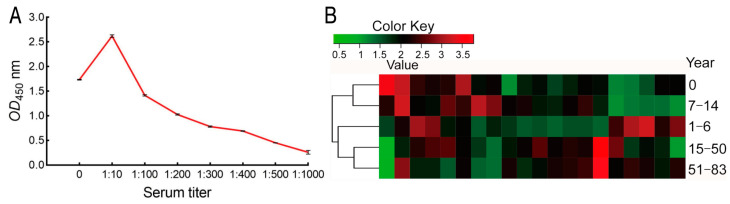
Titer of serum antibodies against outer membrane proteins. (**A**) Selection of desirable serum dilution for detection of serum titer against outer membrane proteins. (**B**) Distribution of serum titer in different stages of age and cluster.

**Figure 5 ijms-25-07450-f005:**
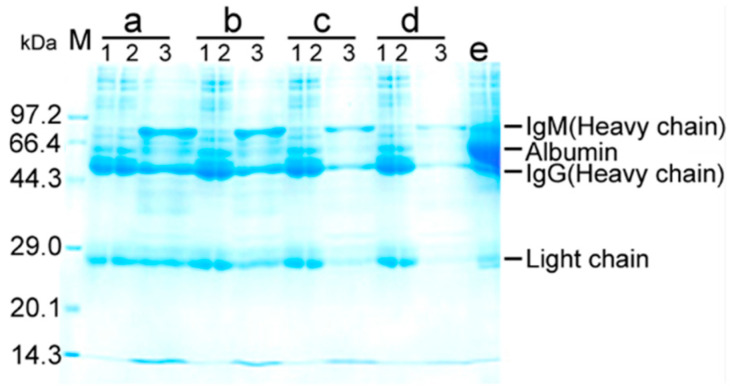
SDS-PAGE analysis for antibody that binds with *E. coli* by bacterial pull-down for percentage of antibodies against *E. coli* in serum immunoglobulin. Pooled sera were purified by 40% saturated ammonium sulfate for immunoglobulins. Aliquots of 32, 24, 16, and 8 mL purified immunoglobulins were mixed with 10^11^ CFU of *E. coli*. After centrifugation, pellets were collected (supernatants were unbinding immunoglobulins). The pellets (the binding of cells and immunoglobulins) were dissociated with 1.2 mL of 0.1 M Gly-HCl (pH 2.4). The supernatant was precipitated overnight at −20 °C with acetone and dissolved in 60 μL PBS. Purified, unbinding, acetone-precipitated immunoglobulins were used for SDS-PAGE analysis. a-1, purified immunoglobulins (5 µL), a-2, unbinding immunoglobulins (5 µL), a-3, binding immunoglobulins (30 µL); b-1, purified immunoglobulins (3.75 µL), b-2, unbinding immunoglobulins (3.75 µL), b-3, binding immunoglobulins (30 µL); c-1, purified immunoglobulins (2.5 µL), c-2, unbinding immunoglobulins (2.5 µL), c-3, binding immunoglobulins (30 µL); d-1, purified immunoglobulins (1.25 µL), d-2, unbinding immunoglobulins (1.25 µL), d-3, binding immunoglobulins (30 µL); e, unpurified antibody by 40% saturated ammonium sulfate.

**Table 1 ijms-25-07450-t001:** Distribution of serum samples in different stages of healthy individuals.

Name	Newborn	Young Children	Children	Youth	Older
Age grades (years)	0	1–6	7–14	15–50	51–83
Samples (portion)	29	30	24	29	29

Note: “0 years” indicates umbilical cord serum.

**Table 2 ijms-25-07450-t002:** Frequency of antibodies against 69 outer membrane proteins in healthy individuals of different stages of age.

*E. coli* Protein	Newborn % (No)	Young Child % (No)	Child % (No)	Youth % (No)	Older % (No)	Total % (No)	SD
BlC	10.34(3)	0(0)	20.83(5)	6.90(2)	3.45(1)	7.80(11)	7.15
CirA	3.45(1)	3.33(1)	4.17(1)	0(0)	0(0)	2.13(3)	1.81
CusC	27.59(8)	33.33(10)	20.83(5)	10.34(3)	6.90(2)	19.86(28)	10.01
CutF	17.24(5)	20.00(6)	29.17(7)	27.59(8)	17.24(5)	21.99(31)	5.13
FadL	0(0)	0(0)	0(0)	0(0)	0(0)	0(0)	0
FaeD	34.48(10)	26.67(8)	33.33(8)	24.14(7)	24.14(7)	28.37(40)	4.48
FecA	0.00(0)	0.00(0)	0(0)	0(0)	0(0)	0(0)	0
FepA	86.21(25)	63.33(19)	95.83(23)	93.10(27)	79.31(23)	82.98(117)	11.63
FhuA	0(0)	3.33(1)	0(0)	0(0)	0(0)	0.71(1)	1.33
FhuE	0(0)	0(0)	0(0)	0(0)	0(0)	0(0)	0
FimD	27.59(8)	10.00(3)	58.33(14)	31.03(9)	20.69(6)	28.37(40)	16.1
FlgH	34.48(10)	56.67(17)	45.83(11)	27.59(8)	58.62(17)	44.68(63)	12.13
Flu	44.83(13)	30.00(9)	37.50(9)	34.48(10)	37.93(11)	36.88(52)	4.85
HlpA	96.55(28)	90.00(27)	100.00(24)	96.55(28)	68.97(20)	90.07(127)	11.2
HofQ	0(0)	0(0)	4.17(1)	0(0)	3.45(1)	1.42(2)	1.88
HtrE	6.90(2)	3.33(1)	8.33(2)	10.34(3)	6.90(2)	7.09(10)	2.29
LamB	0(0)	0(0)	0(0)	0(0)	0(0)	0(0)	0
LolB	0(0)	13.33(4)	0(0)	0(0)	0(0)	2.84(4)	5.33
Lpp	0(0)	0(0)	0(0)	0(0)	0(0)	0(0)	0
MdtP	10.34	0(0)	25	6.9	10.34	9.93	8.17
MipA	86.21(25)	70.00(21)	79.17(19)	75.86(22)	79.31(23)	78.01(110)	5.27
MltA	27.59(8)	43.33(13)	66.67(16)	31.03(9)	44.83(13)	41.84(59)	13.74
MltB	3.45(1)	0(0)	16.67(4)	0(0)	10.34(3)	5.67(8)	6.5
NanC	0(0)	0(0)	4.17(1)	3.45(1)	6.90(2)	2.84(4)	2.64
NfrA	24.14(7)	23.33(7)	20.83(5)	10.34(3)	6.90(2)	17.02(24)	7.1
NlpB	6.90(2)	0(0)	0(0)	3.45(1)	0(0)	2.13(3)	2.76
NmpC	20.69(6)	13.33(4)	25.00(6)	24.14(7)	48.28(14)	26.24(37)	11.74
OmpA	100(29)	96.67(29)	95.83(23)	96.55(28)	100(28)	97.87(138)	2.02
OmpC	0(0)	0(0)	0(0)	0(0)	0(0)	0(0)	0
OmpF	6.90(2)	0(0)	16.67(4)	0(0)	20.69(6)	8.51(12)	8.51
OmpG	10.34(3)	10.00(3)	54.17(13)	31.03(9)	37.93(11)	27.66(39)	16.89
OmpN	75.86(22)	76.67(23)	83.33(20)	62.07(18)	62.07(18)	71.63(101)	8.51
OmpP	6.90(2)	0(0)	8.33(2)	0(0)	0(0)	2.84(4)	3.76
OmpT	0(0)	0(0)	4.17(1)	0(0)	0(0)	0.71(1)	1.67
OmpW	44.83(13)	33.33(10)	45.83(11)	34.48(10)	17.24(5)	34.75(49)	10.32
OmpX	100.00(29)	96.67(29)	100.00(24)	79.31(23)	93.10(27)	93.62(132)	7.69
OstA	13.79(4)	16.67(5)	25.00(6)	34.48(10)	37.93(11)	25.53(36)	9.49
PaL	31.03(9)	33.33(10)	50.00(12)	20.69(6)	34.48(10)	33.33(47)	9.41
PhoE	86.21(25)	86.67(26)	75.00(18)	55.17(16)	62.07(18)	73.05(103)	12.67
PldA	41.38(12)	16.67(5)	62.50(15)	51.72(15)	65.52(19)	46.81(66)	17.64
RzoD	27.59(8)	0(0)	20.83(5)	17.24(5)	24.14(7)	17.73(25)	9.62
Slp	44.83(13)	46.67(14)	29.17(7)	41.38(12)	20.69(6)	36.88(52)	10
SlyB	79.31(23)	90.00(27)	75.00(18)	82.76(24)	68.97(20)	79.43(112)	7.09
TolC	0(0)	0(0)	4.17(1)	3.45(1)	3.45(1)	2.13(3)	1.83
TsX	100.00(29)	96.67(29)	95.83(23)	79.31(23)	93.10(27)	92.91(131)	7.18
VacJ	3.45(1)	3.33(1)	12.50(3)	3.45(1)	6.90(2)	5.67(8)	3.56
WzA	0(0)	0(0)	0(0)	6.90(2)	0(0)	1.42(2)	2.76
YaeT	20.69(6)	10.00(3)	29.17(7)	27.59(8)	17.24(5)	20.57(29)	7.01
YbgQ	31.03(9)	3.33(1)	29.17(7)	24.14(7)	31.03(9)	23.40(33)	10.51
YbhC	68.97(20)	73.33(22)	66.67(16)	65.52(19)	72.41(21)	69.50(98)	3.07
YbiL	62.07(18)	66.67(20)	45.83(11)	51.72(15)	62.07(18)	58.16(82)	7.69
YbjP	79.31(23)	33.33(10)	50.00(12)	55.17(16)	62.07(18)	56.03(79)	15.04
YcbS	44.83(13)	40.00(12)	58.33(14)	48.28(14)	41.38(12)	46.10(65)	6.55
YccZ	0(0)	0(0)	0(0)	0(0)	6.90(2)	1.42(2)	2.76
YedS	0(0)	0(0)	0(0)	3.45(1)	0(0)	0.71(1)	1.38
YehB	0(0)	0(0)	0(0)	6.90(2)	0(0)	1.42(2)	2.76
Yejo	44.83(13)	20.00(6)	62.50(15)	55.17(16)	79.31(23)	51.77(73)	19.7
YfcU	0(0)	0(0)	0(0)	0(0)	0(0)	0(0)	0
YfgL	0(0)	0(0)	0(0)	0(0)	3.45(1)	0.71(1)	1.38
YfiO	10.34(3)	20.00(6)	8.33(2)	0(0)	3.45(1)	8.51(12)	6.83
YhcD	10.34(3)	3.33(1)	41.67(10)	24.14(7)	17.24(5)	18.44(26)	13.14
YiaT	3.45(1)	0(0)	0(0)	10.34(3)	6.90(2)	4.26(6)	4.02
YieC	10.34(3)	20.00(6)	4.17(1)	13.79(4)	17.24(5)	13.48(19)	5.53
YohG	0(0)	0(0)	0(0)	0(0)	0(0)	0(0)	0
YohH	3.45(1)	3.33(1)	0(0)	0(0)	0(0)	1.42(2)	1.66
YpjA	3.45(1)	10.00(3)	8.33(2)	3.45(1)	3.45(1)	5.67(8)	2.85
YqhH	17.24(5)	16.67(5)	29.17(7)	13.79(4)	27.59(8)	20.57(29)	6.25
YqiG	0(0)	0(0)	0(0)	0(0)	0(0)	0(0)	0
YraJ	0(0)	0(0)	0(0)	0(0)	0(0)	0(0)	0

**Table 3 ijms-25-07450-t003:** Active immune protection of outer membrane proteins against challenging by virulent *E. coli* in a mouse model.

Group	Survival	Death	Death Rates (%)	Protective Rates (%)
Control (PBS)	4	11	0.73	----
HlpA	11	4	0.27	63.64 *
TsX	11	4	0.27	63.64 *
OmpA	10	5	0.33	54.54 *
FepA	5	10	0.67	9.09
OmpX	4	11	0.73	0
OstA	13	2	0.13	81.82 *
NlpB	11	4	0.27	63.64 *
OmpC	11	4	0.27	63.64 *
YfcU	11	4	0.27	63.64 *
OmpW	9	6	0.4	45.45
TolC	9	6	0.4	45.45
WzA	9	6	0.4	45.45
NanC	8	7	0.47	36.36
MdtP	7	8	0.53	27.27
PldA	7	8	0.53	27.27
VacJ	7	8	0.53	27.27
YbhC	7	8	0.53	27.27
YedS	6	9	0.6	18.18
YiaT	6	9	0.6	18.18
OmpN	5	10	0.67	9.09
BlC	4	11	0.73	----
FhuE	3	12	0.8	----

* *p* < 0.05 (Compared with PBS control).

**Table 4 ijms-25-07450-t004:** Active immune protection of outer membrane proteins against challenging by *S. aureus* in Kunming mice.

Group	Survival	Death	Death Rate (%)	Protective Rate (%)
Control (PBS)	5	30	77.14	----
HlpA	16	19	45.71	36.67 *
OmpA	22	23	48.57	23.33 *
TsX	4	31	80	----

* *p* < 0.05 (compared with PBS control).

## Data Availability

The data that support the findings of this study are available from the corresponding author, H.L.

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
