# Peer review of "Escherichia coli Activate Extraintestinal Antibody Response and Provide Anti-Infective Immunity"

_ijms, 2024, doi:10.3390/ijms25137450_

Round 1

Reviewer 1 Report

Comments and Suggestions for Authors

It is possible that antigens from intestinal microflora may activate immune response and play a role against infections. This paper indicates that antibodies against E. coli outer membrane proteins are widely present in serum in healthy individuals of different ages,  and there is even the possibility of cross resistance to S. aureus infection. The results are important for exploring the significance of gut microbiota.

Q1. In line 49, the authors gave gut-brain a "", why not gut-lung ? 

Q2. In Fig 2 C, the size of the marked letters is not reasonable, and there are two "Year" in the upper right, it is inexplicable.

Q3. There is no age information labeled in Figure 3B, so what is the difference between the significance of this cluster analysis and that of Figure 2D?

Q4. No error bars shown in Figure 4A. It seems the experiment done only once lacking statistical significance.

Q5. Figure 5B and 5C(table actually)  are not suitable present in the text and should be placed in the supplementary part.

Q6. Because HlpA and OmpA are not components of S. aureus, it is difficult to explain this cross-protective effect. In order to confirm the protective effect of E. coli antibodies against S. aureus infection, it is necessary to perform in vitro immunosorbent assay with HlpA/OmpA immuned mice serum and S. aureus outer membrane protein. Or, at least provide sufficient documentary evidence to demonstrate homology between HlpA/OmpA and S. aureus components.

Comments on the Quality of English Language

The two titles “2.4. Measurement of Serum Antibodies Against E. coli” and “2.5. Quantification of Serum Antibodies Against E. coli” are vague and indistinguishable.

Author Response

Comments 1. In line 49, the authors gave gut-brain a "", why not gut-lung ? 

Response 1, Thank you for your comment. ”” has been added.

Comments 2. In Fig 2 C, the size of the marked letters is not reasonable, and there are two "Year" in the upper right, it is inexplicable.

Response 2, Thank you for your comment. It has been removed.

Comments 3. There is no age information labeled in Figure 3B, so what is the difference between the significance of this cluster analysis and that of Figure 2D?

Response 3, Thank you for your comment. These have explained in lines 166 - 167 as following “Note that this cluster analysis is not consistent with the above cluster (Figure 2D) for these five antibodies due to based antibody title instead of age.”

Comments 4. No error bars shown in Figure 4A. It seems the experiment done only once lacking statistical significance.

Response 4, Thank you for your comment. The experiment has been repeated and the SD have been added.

Comments 5. Figure 5B and 5C(table actually)  are not suitable present in the text and should be placed in the supplementary part.

Response 5, Thank you for your comment. They have been removed into Supplementary Figures.

Comments 6. Because HlpA and OmpA are not components of S. aureus, it is difficult to explain this cross-protective effect. In order to confirm the protective effect of E. coli antibodies against S. aureus infection, it is necessary to perform in vitro immunosorbent assay with HlpA/OmpA immuned mice serum and S. aureus outer membrane protein. Or, at least provide sufficient documentary evidence to demonstrate homology between HlpA/OmpA and S. aureus components.

Response 6, Thank you for your comment. Additional analysis has been added as Supplementary Figure 2 and these description has been added in lines 236-239 as following ” To understand the cross immune protection, phylogenetic trees were performed based amino acid sequence of OmpA and HlpA. OmpA of E. coli and S. aureus was clustered together, while HlpA of E. coli and S. aureus had also a relation (Supplementary Figure 2), suggesting that there is a homology between the two OmpA and HlpA .” and in lines as following “The present study further shows that E. coli OmpA and HlpA provide the cross-protection against S. aureus infection, which is related to the homologies between E. coli OmpA and HlpA and S. aureus OmpA and HlpA, respectively. Therefore, the homology of amino acid sequences provides an explanation for the cross protection.”.

Comments on the Quality of English Language

The two titles “2.4. Measurement of Serum Antibodies Against E. coli” and “2.5. Quantification of Serum Antibodies Against E. coli” are vague and indistinguishable.

Response, Thank you for your comment. They have been modified in line 174 “2.4. Titers of Serum Antibodies against E. coli Outer Membrane Proteins” and line 193 “2.5. Total Serum Antibodies against E. coli Outer Membrane Proteins”.

Reviewer 2 Report

Comments and Suggestions for Authors

In this study, authors analyzed the antibodies to 69 outer membrane proteins of E.coli in 141 healthy individuals. This is unique and interesting study. Study design was elaborated and appropriate methods were used. Though the manuscript is well written and almost perfect, following points should be considered for revision.

1. line 74, and other portions: "gram-negative" should read "Gram-negative". In other parts of main text, there are some words to be corrected as "Gram-positive".

2. In Tables and figures, a number of names of proteins (69) are shown. However, these are difficult to understand for readers. It is recommended to prepare and add a supplementary Table to the revised manuscript. This is a list of the 69 proteins, showing the abbreviated designations, full name (if there is), and other characteristics of protein (if there is; any annotation in GenBank database is enough. It is preferable to show also genomic position of each protein, in any standard E. coli strain.

3. Table 2: leftmost column, please add a head "E. coli protein".

4. section 2.2, 2.4, an other portions. Descriptions of age range should be revised. "0-6 y", "7-14 y", "15-50 y" ... are more commonly used and easy to read. Please see other published papers and follow appropriate description. 

 5. line 298- ; Species names of all the bacteria/fungus should be italicized. 

6. line 293-: Antibodies to OmpA and OmpC display 100% and 0%. For this result, authors discussed differential exposure. How about the positions of outer membrane proteins? Some proteins may be exposed largely to outside the calls, but some may be buried in the membrane with only a small portion or none being exposed outside. In addition how is the relative amount  of outer membrane protein produced?  If large amount is present in the cell membrane, such protein may be easily recognized by immune system of host. If these are not discussed in original version, and if any relevant findings were known, such discussion should be added.   

Author Response

Comments 1: line 74, and other portions: "gram-negative" should read "Gram-negative". In other parts of main text, there are some words to be corrected as "Gram-positive".

Response 1, Thank you for your comment. These have been modified.

Comments 2: In Tables and figures, a number of names of proteins (69) are shown. However, these are difficult to understand for readers. It is recommended to prepare and add a supplementary Table to the revised manuscript. This is a list of the 69 proteins, showing the abbreviated designations, full name (if there is), and other characteristics of protein (if there is; any annotation in GenBank database is enough. It is preferable to show also genomic position of each protein, in any standard E. coli strain.

Response 2, Thank you for your comment. It has been added as supplementary Table 1.

Comments 3: Table 2: leftmost column, please add a head "E. coli protein".

Response 3, Thank you for your comment. These have been added.

Comments 4: section 2.2, 2.4, an other portions. Descriptions of age range should be revised. "0-6 y", "7-14 y", "15-50 y" ... are more commonly used and easy to read. Please see other published papers and follow appropriate description. 

Response 4, Thank you for your comment. These have been modified.

Comments 5: line 298- ; Species names of all the bacteria/fungus should be italicized. 

Response 5, Thank you for your comment. These have been modified.

Comments 6: line 293-: Antibodies to OmpA and OmpC display 100% and 0%. For this result, authors discussed differential exposure. How about the positions of outer membrane proteins? Some proteins may be exposed largely to outside the calls, but some may be buried in the membrane with only a small portion or none being exposed outside. In addition how is the relative amount  of outer membrane protein produced?  If large amount is present in the cell membrane, such protein may be easily recognized by immune system of host. If these are not discussed in original version, and if any relevant findings were known, such discussion should be added. 

Response 6, Thank you for your comment. These have been added in Discussion in lines 304-306 as following “Furthermore, the positions of outer membrane proteins and the relative amount of outer membrane protein produced may be also responsible for it.”

Reviewer 3 Report

Comments and Suggestions for Authors

I suggest more cited referwnces such as:

Harald Brussow. Problems with the voncept of gut microbiota dysbiossis. Doi:10.1111/1751-7915.13479

Tomas Hrncir. Gut microbiota dysbiossis: Triggers, consequences, diagnostic and therapeutic options. Doi: 10.3390/microorganisms10030578

Zugravu et all. Beer and microbiota. Doi:10.3390/nu15040844.

I also suggest to check Englis language, for exampleyou should write Gram with capital because is the name of a Danish MD.

Also replace n with m in Gram, line 74.

Material and metods are well described.

Research design is chosen good.

Results are clearly presented, I suggest a bit of organising the tables to be easily interpreted.

Diecussion section must be extended.

Conclussions support the results.

Comments on the Quality of English Language

There are letters missi g

Line 74 for example

Check all manuscript.

Author Response

Comment 1: I suggest more cited referwnces such as:

Harald Brussow. Problems with the voncept of gut microbiota dysbiossis. Doi:10.1111/1751-7915.13479

Tomas Hrncir. Gut microbiota dysbiossis: Triggers, consequences, diagnostic and

Zugravu et all. Beer and microbiota. Doi:10.3390/nu15040844.

Response 1, Thank you for your comment. These have been cited.

Comment 2: I also suggest to check Englis language, for example you should write Gram with capital because is the name of a Danish MD.

Response 2, Thank you for your comment. These have been modified.

Comment 3: Also replace n with m in Gram, line 74.

Response 3, Thank you for your comment. These have been modified.

Comment 4: Material and methods are well described.

Research design is chosen good.

Results are clearly presented, I suggest a bit of organising the tables to be easily interpreted.

Response 4, Thank you for your comment. An additional Table has been added as Supplementary Table 1.

Comment 5: Diecussion section must be extended.

Response 5, Thank you for your comment. These have been extended in lines 267-269 following as “Therefore, E. coli as a bacterium of intestinal microflora can stimulate extraintestinal acquired immunity to protect hosts from bacterial infection. The finding provides a new insight into the role of the intestinal microflora.”, in lines 304-306 “Furthermore, the positions of outer membrane proteins and the relative amount of outer membrane protein produced may be also responsible for it.”, and in lines 322-325 “The present study further shows that E. coli OmpA and HlpA provide the cross-protection against S. aureus infection, which is related to the homologies between E. coli OmpA and HlpA and S. aureus OmpA and HlpA, respectively. Therefore, the homology of amino acid sequences provides an explanation for the cross protection.”

Conclussions support the results.

Comments on the Quality of English Language

Comment 6: There are letters missi g

Line 74 for example

Response 6, Thank you for your comment. These have been modified.

Comment 7: Check all manuscript.

Response 7, Thank you for your comment. Manuscript has been carefully checked.